# Positive Body Image and Psychological Wellbeing among Women and Men: The Mediating Role of Body Image Coping Strategies

**DOI:** 10.3390/bs14050378

**Published:** 2024-04-30

**Authors:** Camilla Matera, Chiara Casati, Monica Paradisi, Cristian Di Gesto, Amanda Nerini

**Affiliations:** 1Department of Education, Languages, Intercultures, Literatures and Psychology, University of Florence, 50135 Florence, Italy; monica.paradisi@unifi.it (M.P.); cristian.digesto@unifi.it (C.D.G.); amanda.nerini@unifi.it (A.N.); 2School of Psychology, University of Florence, 50135 Florence, Italy; chiara.casati@edu.unifi.it

**Keywords:** positive body image, body appreciation, body functionality, body compassion, coping strategies, self-acceptance, wellbeing

## Abstract

This study aimed to examine the mediating role of body image coping strategies in the relationship between positive body image and wellbeing. Three hundred and seventy-two women and three hundred and seventy-seven men completed a questionnaire assessing body appreciation, body appreciation functionality, body compassion, body image coping strategies (appearance fixing, avoidance, positive rational acceptance), self-acceptance and overall psychological wellbeing. Path analysis showed that avoidance significantly mediated the relationship between body appreciation and overall psychological wellbeing among both women and men; its mediating role was confirmed for men’s but not for women’s self-acceptance. Positive rational acceptance was a significant mediator of the relationship between body compassion and both psychological wellbeing and self-acceptance among men but not among women. These findings show that higher body appreciation is associated with a lower tendency to avoid appearance-related cognitions or thoughts that are interpreted as threatening, with an indirect effect on women’s and men’s psychological wellbeing. Analogously, but only for men, body compassion is associated with mental activities and self-care behaviors that foster rational self-talk and the acceptance of one’s experiences, which, in turn, are linked to higher wellbeing. These findings can help to plan programs aimed at fostering individuals’ wellbeing by focusing on their positive body image considering gender differences.

## 1. Introduction

Although prevalent among women [1,2], dissatisfaction with one’s physical appearance is growing also among men [3,4]. Body dissatisfaction is very common in Western societies, such as Italy. In a study conducted in 24 European countries, USA and Canada, Italian male adolescents reported the highest rates of body dissatisfaction (39.9%); body dissatisfaction in the Italian female sample reached 55.2%, a very high percentage, surpassed only by two other countries [5]. These findings are in line with the ones reported by Jaeger et al. [6], who found that, compared to other countries, Northern Italy had among the lowest BMI rates and the highest rates of overestimation of one’s body shape. A different type of body dissatisfaction is observed between genders in Italy: girls tend to aspire towards thinness, while boys desire increased body mass [7,8], although a thin body is also valorized [9].

In Western countries, both women and men are frequently exposed to an abundance of images that promote near-perfect bodies [10,11,12], so they are likely to experience threats to their body image [13]. Indeed, several common situations or events (e.g., looking at oneself in the mirror, wearing a swimsuit, looking at images posted on social networks) might trigger body-related thoughts and feelings that are experienced as unwanted and oppressing [13]. To cope with them, individuals tend to employ cognitive and behavioral strategies [14]. Drawing on the conceptualization of positive body image, this study aimed to examine if the ability to appreciate and celebrate, rather than devalue, one’s body is associated with body image coping strategies, which in turn can have positive effects on both women’s and men’s wellbeing.

### 1.1. Body Image Coping Strategies

Cash et al. [14] described three strategies that can be used to cope with body-related threats: positive rational acceptance, avoidance and appearance fixing. Positive rational acceptance involves mental activities and self-care behaviors that foster rational self-talk and the acceptance of one’s experiences by refocusing on personal resources [14]. Avoidance involves shunning confrontation with a feared situation that makes one’s body image salient or avoiding appearance-related cognitions or thoughts that are interpreted as threatening. Appearance fixing consists of making changes and corrections in one’s outward appearance to try to camouflage or mask physical features perceived as inadequate [14]. Both avoidance and appearance fixing are described as maladaptive coping strategies [14] because they constitute a negative reinforcement for those cognitive processes underlying body dissatisfaction and body image distress by temporarily reducing aversive appearance-related cognitions and emotions [13]. Conversely, positive rational acceptance is considered adaptive because it promotes a more balanced and compassionate view of oneself, allowing individuals to focus on their strengths and personal values [13].

Body image and coping strategies are key elements for one’s wellbeing. Indeed, research has shown that adopting certain coping strategies might be related to specific mental health outcomes, such as eating disorders symptoms [14]; body shame [15]; bulimic, depressive and anxiety symptoms; drive for thinness [16]; greater body image inflexibility [17]; lower life satisfaction and self-esteem; higher negative affect; and body surveillance [18].

Cash et al. [14] found that the three coping strategies were significant predictors of both body satisfaction and disturbed eating attitudes and behaviors among North American young women. Among men, avoidance and positive rational acceptance significantly predicted body satisfaction, while avoidance and appearance fixing predicted disturbed eating attitudes and behaviors.

In a mixed sample of Australian teenagers, Hughes and Gullone [16] observed that avoidance predicted bulimic, depressive and anxiety symptoms and drive for thinness; positive rational acceptance predicted depressive symptoms only, while appearance fixing predicted drive for thinness only. Moreover, avoidance was found to moderate the relationship between body image concerns and bulimic symptoms, whereas positive rational acceptance moderated the link between body image concerns and depressive symptoms.

In a study involving Australian women aged 18-51, Mancuso [17] investigated the relationship between body image evaluation and maladaptive strategies through body image inflexibility. The results showed that appearance evaluation had a significant effect on both appearance-fixing and avoidance behaviors via body image inflexibility.

In two independent samples of Canadian women, Choma et al. [15] found that body shame predicted greater use of appearance fixing and avoidance coping rather than less reliance on positive rational acceptance. The relationship between body shame and subjective wellbeing was partially mediated by avoidance coping but not by the other coping strategies. The role of body image coping strategies as moderating variables was not confirmed.

Rollero et al. [18] examined the three coping strategies in a sample of Italian men and women. They found that positive rational acceptance was not significantly associated with either body shame, body surveillance or indices of wellbeing (such as life satisfaction, positive and negative affect, self-esteem), while both appearance fixing and avoidance were significantly associated with higher body shame and body monitoring, lower self-esteem and higher negative affect. Avoidance also displayed negative associations with life satisfaction and positive affect.

Taken together, these findings suggest that coping strategies, especially maladaptive ones, are significantly linked to several indicators of health and subjective wellbeing. Nevertheless, we can observe that only little attention has been devoted to the link between body image coping strategies and psychological wellbeing. According to Ryff [19], wellbeing cannot be considered as the mere presence of positive affect or satisfaction with life, but it is constituted by a sense of living a purposeful life which is also meaningful and characterized by the presence of positive relationships with others (see also [20]). In this view, wellbeing can be defined as the pursuit of the realization of one’s true potential, which is articulated in different dimensions: autonomy, personal growth, self-acceptance, purpose in life, positive interpersonal relationships and environmental mastery. Among them, self-acceptance, which can be defined as a positive attitude toward oneself, is especially relevant when considering individuals’ self-view with respect to their bodies. Self-acceptance, which relates to being aware of and able to accept one’s strengths and weaknesses, is a central characteristic of positive psychological functioning and mental health, as well as a characteristic of self-actualization, optimal functioning and maturity [19]. The lack of capacity to accept oneself while instead focusing on self-evaluation can result in some negative emotional outcomes, such as depression [21], uncontrolled anger [22] and eating disorder behaviors (e.g., dietary restraint, compensatory behaviors, binge eating) [23].

Moreover, previous research concerning body image coping strategies focused on negative body image, while very little is known about the link between positive body image and body image coping strategies.

### 1.2. Positive Body Image

Positive body image has been defined as an overarching love, acceptance and respect for the body, including its imperfections and features that are not in line with idealized standards of beauty [24]. Positive body image is a multifaceted construct that is not on the same continuum as negative body image. It permits people to appreciate the unique beauty and functions of their body, to which they might feel connected in a mindful way; it allows them to interpret incoming information in a protective rather than threatening manner [24,25]. If individuals accept and value their bodies, they may be less concerned with changing their appearance, and they might focus more on how their bodies feel and function rather than on how they appear [24]. Positive body image is associated with many indices of wellbeing, e.g., [26,27,28], although further research is needed to identify variables that can be responsible for this link.

Positive body image includes a series of constructs, such as appreciation of the unique beauty of the body, appreciation of one’s body functionality and body compassion [24,25,29]. Body appreciation is defined as an intentional choice to accept one’s own body regardless of its appearance, to respect and take care of it and its needs through routines and behaviors that promote wellness, refusing the unrealistic ideals of appearance proposed by the media [30]. A positive linear relationship has been found between age and body appreciation, with older women reporting higher levels of body appreciation than younger women [28,31,32]. Some studies showed that body appreciation is higher among men [28,33,34,35,36], while others revealed higher body appreciation levels among women [37] or found no gender differences [34].

Appreciating one’s body functionality helps individuals refocus the way they think of their body, not paying attention to its imperfections but to its functions and capabilities [38]. Some studies suggest that young women are more oriented towards appearance [38,39], whereas as the years go by, the attention shifts toward one’s physical functionality [32,40,41]. However, a recent meta-analysis [42] did not find evidence of age-related differences for either women or men. Moreover, results concerning gender differences are inconsistent [42,43].

Body compassion refers to kind and comprehensive behaviors toward personal delusions and difficulties regarding one’s own body [44]. This construct is tied to mindfulness and acceptance behaviors while incorporating a multidimensional definition of body image [45]. Research on body compassion is relatively new and in continuous evolution. Studies on the effect of age and gender are few and non-exhaustive [46,47,48].

### 1.3. Positive Body Image and Coping Strategies

Little research has been conducted on the relationship between positive body image and coping strategies. Some exceptions are represented by few studies conducted to test the effectiveness of programs for positive body image enhancement [49,50,51]. Regehr et al. [51] showed that adolescents who had participated in the positive body image program “Free to be” had an improvement in body image coping strategies from pre-test to post-test, although no significant differences emerged with participants in the control condition. According to the authors, by increasing positive body image, participants learned strategies for coping with body image threats, but not to the extent it was expected. In testing the effectiveness of group therapy sessions focusing on “Beautiful body image”, Irani et al. [49] found that Iranian adolescents who took part in the group activity exhibited increased positive rational acceptance coping strategies and higher satisfaction with their own bodies, compared to the control group. Avoidance and appearance fixing did not vary between the intervention and control groups. Keven-Akliman and Eryilmaz [50] found a significant increase in Turkish women’s body satisfaction and positive rational acceptance in individuals taking part in positive psychotherapy sessions compared to the control group. Even if not immediately after the intervention, a significant decrease in maladaptive coping strategies was observed in the six-month follow-up, when participants in the experimental group also kept on presenting higher positive rational acceptance compared to participants in the control group. Although limited, this literature seems to indicate that positive body image is associated with the implementation of adaptive coping strategies at the expense of maladaptive ones.

### 1.4. The Present Study

As we have seen, research on the association between positive body image and body image coping strategies is scarce and leaves several issues unresolved. First, previous research has not distinguished between different facets of positive body image. Second, the studies assessing the link between positive body image and coping strategies have mainly focused on adolescents [49,52]. The only study conducted on adults involved female but not male participants. Third, it is not clear if positive body image might contribute to individuals’ wellbeing via body image coping strategies, as the link between these factors has never been investigated before. Fourth, many indices of wellbeing have been considered, but little attention has been devoted to the link between body image coping strategies and psychological wellbeing.

Based on these considerations, the present study aimed to explore the link between positive body image, in terms of body appreciation, body functionality appreciation and body compassion, as well as both women’s and men’s psychological wellbeing via maladaptive and adaptive coping strategies. Besides considering overall psychological wellbeing, we focused on self-acceptance, which is a core dimension of individuals’ psychological functioning, especially in the context of body image [21].

It is our contention that individuals who can appreciate and praise their body beyond its flaws and inadequacies should be less likely to experience external stimuli as threats to their body image and consequently should resort less frequently to maladaptive coping strategies, such as avoidance and appearance fixing. Conversely, they should be more accustomed to self-reassurance by implementing positive rational acceptance coping. In turn, avoidance and appearance fixing might be related to lower levels of wellbeing and self-acceptance, while relying on positive rational acceptance might be associated with higher psychological wellbeing and higher self-acceptance.

Based on these considerations, we developed a theoretical model in order to test some specific hypotheses.

**Hypothesis 1 (H1).** 
*Higher body appreciation, functionality appreciation and body compassion were hypothesized to be associated with higher levels of positive rational acceptance and lower levels of avoidance and appearance fixing.*


**Hypothesis 2 (H2).** 
*Avoidance and appearance fixing were supposed to be negatively associated with psychological wellbeing and self-acceptance, while positive rational acceptance was hypothesized to be a positive predictor of these outcome variables.*


**Hypothesis 3 (H3).** *The three coping strategies were hypothesized to partially mediate the relationship between positive body image (in terms of body appreciation, functionality appreciation and body compassion) and both psychological wellbeing and self-acceptance*. 

Given that previous research highlighted significant associations between age and positive body image and body image coping strategies, we tested our hypotheses and controlled for the effect of this variable. On two independent samples of women and men, we tested two different models, the outcome variable of which was, respectively, overall psychological wellbeing and self-acceptance.

## 2. Materials and Methods

### 2.1. Participants

Participants were 372 women and 377 men. Women had a mean age of 34.79 (SD = 14.63), ranging from 18 to 81 years. Their average BMI was 22.78 (SD = 3.98). Almost all of the women who took part in the study were Italian (97%), and only 3% were of foreign nationality. Regarding civil status, 27.2% were married or cohabitating, 37% had a partner but were not married or cohabitating, 30.9% were without a partner, 3.2% were separated or divorced, 1.1% were widows and 0.5% declared another civil status. Regarding education, 34.7% had a bachelor’s degree, 33.6% had a high school degree, 27.7% had a master’s degree, 3% had completed lower secondary school, and 1% reported a different degree (e.g., Ph.D.). With respect to occupation, 44.9% were students, 28.5% had full-time employment, 13.2% had part-time employment, 4% had occasional work, 4.3% were retired, 2.4% reported that they were looking for their first job, 1.6% were unemployed, and 1.1% were housewives.

Men participants had a mean age of 39.72 (SD = 15.84), ranging from 18 to 83 years, and their average BMI was 25.09 (SD = 3.98), ranging from 16.41 to 43.25. Almost all of the male participants were Italian (98.1%), and only 1.9% were of foreign nationality. Regarding civil status, 34.7% were married or cohabitating, 30.8% had a partner but were not married or cohabitating, 26.8% were without a partner, 6.4% were separated or divorced, 0.8% declared another civil status, and 0.5% were widows. Regarding education, 46.7% had a high school degree, 24.9% had a bachelor’s degree, 19.6% had a master’s degree, 6.6% had completed lower secondary school, and 2.2% reported a different degree (e.g., Ph.D.). With respect to occupation, 56.5% had full-time employment, 26.5% were students, 6.6% had part-time employment, 6.1% were retired, 2.4% had occasional work, and 1.9% were unemployed.

### 2.2. Measures

**Body Appreciation.** The Italian version [52] of the Body Appreciation Scale-2 (BAS-2) [24] was used to assess participants’ appreciation of their bodies. The scale has 10 items (e.g., “I feel good about my body”) rated on a 5-point Likert scale (1 = never; 5 = always). The Italian version proved to be valid for both men and women [52]. High scores indicated greater levels of body appreciation. In the present study, Cronbach’s a of the scale was very good (women: α = 0.94; men: α = 0.92).

**Body Compassion**. The Italian version [53] of the body compassion scale (BCS) [54] was used to assess compassion toward one’s body. The scale is composed of 23 items (e.g., “I am accepting of my looks just the way they are”) rated on a 5-point Likert scale (1 = almost never; 5 = almost always). High scores indicated greater levels of body compassion. The validity and reliability of the Italian version were good for both women and men [53]. In the present study, the scale presented good internal consistency among both women and men (women: α = 0.89; men: α = 0.86).

**Functionality Appreciation**. The Italian version [55] of the Functionality Appreciation Scale [25] was used to assess levels of appreciation, respect and gratitude towards one’s body for what it is capable of doing. The scale is composed of 7 items (e.g., “I appreciate my body for what it is capable of doing”) rated on a 5-point Likert scale (1 = strongly disagree; 5 = strongly agree). The Italian version proved to be valid for both men and women [56]. High scores indicated greater levels of appreciation and respect for the functions of one’s body. In the present, the FAS presented good internal consistency (women: α = 0.87; men: α = 0.88).

**Body Image Coping Strategies**. The Italian version [18] of the Body Image Coping Strategies Inventory (BICSI) [14] was used to assess coping strategies with body image threats and challenges. The BCSI is composed of 24 items, with a 4-point Likert-type response format (0 = definitely not me; 3 = definitely me). Three subscales comprise the BICSI. The appearance fixing subscale measures the attempt to alter image with efforts to disguise, hide, camouflage, or alter the body area that the individual deems undesirable (9 items; e.g., “I think about what I should do to change my looks”). The avoidance subscale measures the extent to which an individual will avert psychological discomfort through self-imposed ignorance of one’s undesirable thoughts or feelings (7 items; e.g., “I tell myself that I am helpless to do anything about the situation”). The positive rational acceptance subscale measures the degree to which people adopt behavioral and mental strategies to pacify distress through positive self-care (8 items; e.g., “I tell myself that I probably look better than I feel that I do”). The Italian version proved to be valid for both men and women [18]. High scores in the BICIS subscales indicated greater levels of appearance fixing, avoidance and positive rational acceptance of body image-related coping strategies, respectively. In the present study, Cronbach’s a of the scales were satisfactory (women: appearance fixing: α = 0.85, avoidance: α = 0.79; positive rational acceptance: α = 0.78; men: appearance fixing: α = 0.85, avoidance: α = 0.76, positive rational acceptance: α = 0.80).

**Psychological wellbeing and self-acceptance**. Participants’ psychological wellbeing was assessed through the Italian short-form version [56] of Ryff’s Psychological Wellbeing Scale (RPWB) [57]. The short RPWB is composed of 18 items rated on a 6-point Likert-like scale (from 1 = definitively disagree to 6 = definitively agree). The internal consistency of the overall scale was good for both women and men (women: α = 0.86; men: α = 0.86). High scores indicate high levels of wellbeing. As outlined in the hypotheses, in addition to the scale’s overall score, we focused on the self-acceptance subscale, which is composed of three items (e.g., “When I look at the story of my life, I am pleased with how things have turned out”). The internal consistency for self-acceptance subscale was good (women: α = 0.80; men: α = 0.82).

### 2.3. Procedure

Participants were invited to take part in an online survey shared through social networking sites (i.e., Facebook groups, Instagram, LinkedIn). Participation in the study was voluntary, and we did not provide any incentive. To be eligible for the study, respondents had to be 18 years or older. We obtained informed consent from each participant prior to administering the questionnaire. The procedures were clearly explained, and participants could interrupt or quit the survey at any time without explanation. The questionnaire was anonymous, did not ask for any personally identifiable information and took about 20 min to complete. The study was part of a larger project that was approved by the Ethical Committee of the University to which the authors were affiliated (Prot. n. 0148845, 5 July 2023).

### 2.4. Data Analysis

All the analyses were conducted separately on men and women. Descriptive statistics (means and standard deviation) and intercorrelations between all the variables were computed. To interpret the extent of the correlation, we refer to Cohen [58], for which an r value between 0.10 and 0.30 in absolute value indicates a small association, between 0.30 and 0.50 indicates a moderate correlation, and greater than 0.50 indicates a strong association. All the assumptions for path analysis were satisfied [59]. Path analysis was performed to test our hypotheses, placing the positive body image variables (body appreciation, body compassion, functionality appreciation) as the predictors, psychological wellbeing and self-acceptance, respectively, as the criterion variable and coping strategies for threats related to body image as the mediator. Path models were analyzed using AMOS 24 [60], and the model parameters were estimated based on bias-corrected estimates derived from 1000 bootstrap samples computed using the maximum likelihood procedure. We tested model fit through several indexes: Chi-Square (χ^2^), χ^2^/df, the Root Mean Square Error of Approximation (RMSEA), the Standardised Root Mean Square Residual (SRMR), the Comparative Fit Index (CFI) and the Incremental Fit Index (IFI). Assessing the ratio between χ^2^ and the degree of freedom is recommended as a measure of model fit since the χ^2^ statistic is sensitive to sample size. A ratio not greater than 5 is considered acceptable [61]. An acceptable model fit would be indicated by CFI and IFI higher than 0.90, RMSEA between 0.08 and 0.10, and SRMR lower than 0.08; a good fit would be indicated by CFI and IFI higher than 0.95, RMSEA lower than 0.08, and SRMR lower than 0.05 [62].

## 3. Results

No missing values were present in the dataset. Table 1 shows the descriptive statistics (means and standard deviations) of the research variables for women and men. As we can see from the Table, all of the data were normally distributed (women: skewness < −1.07; kurtosis < 2.20; men: skewness < −1.28; kurtosis < 2.68), according to conventional criteria (skews lower than 2 and kurtosis lower than 7) [63].

Correlations between all variables are presented in Table 2. Body appreciation shows a negative association with appearance fixing and avoidance coping strategies in both women and men, resulting in being stronger between women rather than men. Moreover, body appreciation is positively associated with positive rational acceptance, with a small correlation in men and a medium one between women. Also, functionality appreciation was associated negatively with avoidance and appearance fixing for both genders, while it was associated positively with positive rational acceptance, with a small correlation in men and a medium one between women. Finally, body compassion reported a similar pattern, with a negative medium association with avoidance and appearance fixing in the women group, while in the men sample, it reported little negative association with the two coping strategies. Regarding positive rational acceptance, a positive medium correlation to body compassion was reported for both genders.

Concerning the association with wellbeing, the three measures of positive body image were positively correlated to both overall wellbeing and self-acceptance in women and men. In particular, while functionality appreciation and body compassion had a medium association, for body appreciation, the correlation was large with both wellbeing and self-acceptance for the two genders. Also, coping strategies were found to be significantly associated with self-acceptance and overall wellbeing in both women and men. In particular, appearance fixing reported little negative associations, while avoidance showed negative medium associations, and finally, positive rational acceptance reported little positive associations with both overall wellbeing and self-acceptance in women and men.

We then tested our theoretical model through path analysis in women and men. As predicted, body appreciation, functionality appreciation and body compassion were significantly intercorrelated, so they were allowed to covary in our models. Moreover, based on the observed correlations, age was allowed to covary with body compassion among men and with both body compassion and body appreciation among women.

We first tested our model with overall psychological wellbeing as the criterion variable among women (Figure 1) and men (Figure 2). Path analysis showed that the hypothesized model fitted quite well among women (χ^2^ = 9.65, *p* = 0.01, χ^2^/df = 4.81, CFI = 0.99, IFI = 0.99, RMSEA = 0.10, SRMR = 0.02) and very well with the data among men (χ^2^ = 9.09, *p* = 0.03, χ^2^/df = 3.03, CFI = 0.99, IFI = 0.99, RMSEA = 0.07, SRMR = 0.03).

We then tested our model with self-acceptance as the criterion variable for women (Figure 3) and men (Figure 4). Path analysis showed that the hypothesized model fitted very well with the data among both women (χ^2^ = 5.37, *p* = 0.07, χ^2^/df = 2.69, CFI = 0.99, IFI = 0.99, RMSEA = 0.07, SRMR = 0.02) and men (χ^2^ = 6.37, *p* = 0.09, χ^2^/df = 2.12, CFI = 0.99, IFI = 0.99, RMSEA = 0.05, SRMR = 0.02).

Hypothesis 1 was partially confirmed among both women and men. Among women, body appreciation was significantly associated with the three coping strategies, so higher levels of body appreciation corresponded to lower levels of avoidance and appearance fixing and higher levels of positive rational acceptance. Among men, similar findings emerged, although the link between body appreciation and positive rational acceptance was not significant. Women’s functionality appreciation was not significantly associated with avoidance, while it was related to appearance fixing and positive rational acceptance. Nevertheless, both these correlations were positive, so that higher appreciation of one’s body functionality corresponded to higher positive rational acceptance and higher, rather than lower, appearance fixing. Among men, functionality appreciation was not significantly related to any coping strategies. In line with our hypothesis, higher body compassion was associated with lower appearance fixing and higher positive rational acceptance among women and men, while it was significantly related to avoidance only among women.

Hypothesis 2 was confirmed with respect to avoidance, which was negatively associated with self-acceptance and overall psychological wellbeing among both women and men, while it was not confirmed with respect to appearance fixing. Among women, contrary to our prediction, the link from appearance fixing to self-acceptance was positive, while the one to overall psychological wellbeing was not significant. Among men, appearance fixing was not significantly related to either self-acceptance or psychological wellbeing. Regarding positive rational acceptance, Hypothesis 2 was confirmed for men, for whom this coping strategy was positively associated with both self-acceptance and overall psychological wellbeing, but not for women, for whom none of these links was significant.

Our mediational hypothesis (Hypothesis 3) was partially confirmed among women, for whom an indirect link from body appreciation to overall wellbeing via avoidance was observed (Table 3).

Nevertheless, when considering self-acceptance as the criterion variable, the indirect links from either body appreciation, body functionality or body compassion were not significant, as zero was included in the 95% confidence interval (Table 4). Hypothesis 3 was partially confirmed among men, for whom a significant indirect effect of body appreciation on both self-acceptance and overall wellbeing was observed, and a significant indirect effect of body compassion on the criterion variables emerged, while the indirect effect of body appreciation was mediated by avoidance, the indirect effect of body compassion was mediated by positive rational acceptance.

Among women, the models accounted for much of the variance in self-acceptance (33%) and psychological wellbeing (37%). An even greater portion of the variance in self-acceptance (33%) and psychological wellbeing (43%) was accounted for among men.

## 4. Discussion

The current study sought to investigate the relationship between positive body image, conceptualized as body appreciation, body functionality appreciation and body compassion, as well as the psychological wellbeing of both women and men through maladaptive (i.e., avoidance and appearance fixing) and adaptive coping strategies (i.e., positive rational acceptance). In addition to examining overall psychological wellbeing, the study also considered self-acceptance, as it represents a fundamental aspect of individuals’ psychological functioning, particularly within the context of body image [21].

Hypothesis 1 was partially supported for both women and men. Among women, higher levels of body appreciation correlated with lower levels of avoidance and appearance fixing, as well as higher levels of positive rational acceptance. Similar patterns were observed among men, but the link between body appreciation and positive rational acceptance was not statistically significant. The significant association between body appreciation and coping strategies in women may be influenced by societal norms and pressures related to body image. Women often face greater societal expectations regarding appearance [64], leading to a stronger connection between body appreciation and coping mechanisms like avoidance and appearance fixing. Conversely, the lack of a significant relationship between body appreciation and positive rational acceptance in men could be related to masculinity norms and gender role expectations. Societal norms often impose different standards for men and women regarding body image and self-acceptance [65]. Men are typically encouraged to prioritize traits like strength and physical fitness, while women are pushed to focus more on appearance and beauty standards [65,66]. These gendered expectations shape individuals’ perceptions of their bodies and influence their coping strategies.

For men, the emphasis on physical strength and appearance ideals may create a gap between body appreciation and positive rational acceptance. Men may tend to value their bodies based on external factors like muscularity or physical fitness rather than inner qualities and self-care behaviors [29,67] that promote positive rational coping strategies. This difference in values and priorities could weaken the link between body appreciation and positive rational acceptance in men compared to women.

In women, there is a unique pattern of associations between functionality appreciation and coping strategies that suggest functionality appreciation plays a distinct role in shaping their responses to challenges. Functionality appreciation was not significantly linked to avoidance but showed positive correlations with appearance fixing and positive rational acceptance. Functionality appreciation reflects how individuals value the functional aspects of their bodies, which can be linked to a sense of agency and self-efficacy in managing body-related challenges [68,69]. Women who appreciate the functionality of their bodies may see themselves as capable and competent in addressing appearance-related issues, leading to a proactive approach rather than avoidance in enhancing their physical presentation [70]. Women valuing the functional aspects of their bodies may prioritize efforts to enhance their appearance to align their external presentation with their internal sense of physical competence and wellbeing. Moreover, by recognizing and accepting their bodies’ functional abilities and requirements, women may be better prepared to approach stressors with a positive and rational mindset, resulting in more effective coping responses [71].

On the other hand, our research revealed a lack of a significant relationship between functionality appreciation and coping strategies among male participants. Societal norms and gender expectations regarding masculinity and body image may influence men’s coping strategies in a way that diminishes the association with body functionality appreciation [69]. Men may be socialized to prioritize outward appearances, muscularity and physical prowess as markers of masculinity, which could overshadow the significance of valuing the functional aspects of their bodies in relation to coping behaviors [72]. This emphasis on appearance norms and societal ideals of masculinity may shift men’s focus on appearance-enhancing coping strategies rather than functional aspects of body image.

In line with our hypothesis, higher levels of body compassion were associated with lower engagement in appearance-fixing behaviors and higher levels of positive rational acceptance in both women and men. Individuals who demonstrate greater self-compassion tend to treat themselves with kindness and understanding, leading to decreased tendencies to excessively focus on appearance concerns and engage in maladaptive coping behaviors [73,74,75]. The significant association between body compassion and avoidance coping strategy, specifically among women, suggests a gender-specific pattern in the utilization of coping mechanisms. Women may be more inclined to use avoidance strategies in response to body image concerns when they possess lower levels of body compassion. This could be influenced by societal pressures, cultural norms and internalized beliefs about body image that may lead women to avoid confronting or addressing body-related issues when they do not exhibit self-compassion [73,76]. Higher body compassion levels, which entail self-acceptance, self-care and kindness towards one’s body, women may prevent women from resorting to avoidance strategies as a means of self-protection in the face of threats to their body image [77].

Hypothesis 2 was supported in relation to avoidance, showing a negative association with self-acceptance and overall psychological wellbeing in both women and men. This finding underscores the maladaptive nature of avoidance behaviors in addressing body image concerns, as they may hinder individuals from cultivating self-acceptance and experiencing positive psychological wellbeing. However, this hypothesis was not supported in terms of appearance fixing. Among women, contrary to our hypothesis, the relationship between appearance fixing and self-acceptance was positive, while the link to overall psychological wellbeing was not statistically significant. This unexpected positive association may suggest that engaging in efforts to improve or change one’s appearance could be viewed as a form of self-care or self-improvement, potentially enhancing feelings of self-acceptance. However, the absence of a significant connection between appearance fixing and overall psychological wellbeing suggests that the advantages of appearance-focused coping strategies may be limited in terms of broader psychological outcomes. Women may excessively rely on this body image coping mechanism without realizing that it does not lead to long-term benefits. Among men, appearance fixing was not significantly related to either self-acceptance or psychological wellbeing. In many cultures, men are less socialized than women to modify their appearance, resorting to make-up, clothes or particular hairstyles [78].

In relation to positive rational acceptance for men, Hypothesis 2 was supported, indicating a positive association between this coping strategy and both self-acceptance and overall psychological wellbeing. However, this relationship was not significant for women. Women may face unique challenges and pressures related to body image, self-acceptance and coping mechanisms due to societal norms that emphasize appearance and beauty standards. These societal expectations may shape how women navigate their self-perceptions and coping strategies, leading to a more nuanced relationship between positive rational acceptance and psychological outcomes. Further research is needed to delve deeper into the specific coping mechanisms and strategies utilized by women in different contexts, considering the multifaceted influences of gender roles and societal expectations on their wellbeing.

Our mediational hypothesis (Hypothesis 3) was partially supported for women, as an indirect path from body appreciation to overall wellbeing through avoidance was identified. However, when examining self-acceptance as the outcome variable, the indirect pathways from body appreciation, body functionality, or body compassion were not found to be statistically significant. This path suggests that individuals who exhibit a greater body appreciation may be more inclined to engage in avoidance strategies as a means of managing stressors related to body image concerns, with negative outcomes on their overall sense of wellbeing. The interplay between body appreciation, functionality and compassion with self-acceptance may be mediated by other psychological processes or contextual variables that were not accounted for in the current analysis. It is plausible that lower levels of body dissatisfaction may have a more significant impact on self-acceptance than high levels of positive body image, given that these two constructs do not represent opposite ends of the same spectrum but rather are relatively independent [24].

Hypothesis 3 was partially supported among men, revealing a significant indirect effect of body appreciation on both self-acceptance and overall wellbeing, as well as a significant indirect effect of body compassion on these criterion variables. The indirect effect of body appreciation was mediated by avoidance, while the indirect effect of body compassion was mediated by positive rational acceptance. These findings can be explained by the interplay of different coping mechanisms in the relationship between body image and psychological outcomes. The ability to appreciate one’s body tends to decrease the use of maladaptive coping strategies in men, while a compassionate attitude tends to activate available self-care resources, thus promoting the utilization of adaptive coping strategies. This aligns with observations regarding self-compassion, which has been associated with the mobilization of internal resources that bolster wellbeing, including self-soothing capabilities, emotional regulation competencies and a positive self-perception [79,80]. By accessing these resources, individuals enhance their capacity to navigate difficult circumstances and effectively manage stressors [79,80]. Further research is needed to explore the underlying mechanisms and individual differences that contribute to the varying impacts of body appreciation and body compassion on self-acceptance and overall wellbeing in men.

Finally, despite evidence of the role of body image coping strategies as mediating factors, positive body image remained a significant direct predictor of psychological wellbeing, confirming its relevant role in helping individuals reach their true potential. While body image coping strategies may play a role in how individuals manage their perceptions of their bodies, positive body image itself is directly linked to psychological wellbeing in a more fundamental way. Individuals with a positive body image are more likely to experience lower levels of anxiety, depression and stress and are better equipped to navigate life’s challenges with resilience and confidence [81]. By directly predicting psychological wellbeing, positive body image may act as a protective factor that fosters a healthy mindset and emotional balance. This sense of contentment with one’s body can contribute to individuals feeling more empowered, capable and able to pursue their goals and aspirations, ultimately helping them reach their full potential in various aspects of their lives [26,81]. Therefore, even in the presence of body image coping strategies, the inherent benefits of cultivating a positive body image seem to have a significant and direct impact on promoting psychological wellbeing, highlighting its importance in supporting individuals in realizing their true capabilities and leading fulfilling lives.

The main contribution of this study was to investigate the relationship between positive body image perceptions and coping strategies, as well as the link between body image coping strategies and psychological wellbeing, topics that have received limited attention in previous research. The study’s strength lies in its utilization of a multidimensional approach to the examined constructs. Specifically, body image was assessed through the perspectives of body appreciation, functionality appreciation and body compassion, encompassing both maladaptive strategies (e.g., avoidance and appearance fixing) and adaptive strategies (e.g., positive rational acceptance). Moreover, the study’s incorporation of various outcome measures, such as overall psychological wellbeing and self-acceptance, facilitates a comprehensive comprehension of the implications of the study variables.

Furthermore, the study conducted separate analyses for women and men, enabling a nuanced examination of gender-specific differences in the variables and relationships. This gender-specific analysis is crucial for identifying distinct patterns, inequalities and requirements within each gender group, which is essential for devising targeted and efficacious interventions. By elucidating these distinctions, interventions can be tailored to be more specific, pertinent and impactful, thereby enhancing the likelihood of positive outcomes and success in addressing particular challenges or issues.

The tested model explained a substantial portion of the variance in self-acceptance and psychological wellbeing, indicating that the study variables possess significant explanatory power in predicting these outcomes. This underscores the importance and relevance of the variables under scrutiny, suggesting that they are pivotal in shaping individuals’ self-acceptance and psychological wellbeing. Consequently, this knowledge can inform interventions, programs or strategies aimed at fostering positive self-acceptance and psychological wellbeing among individuals in the target population.

Nevertheless, we should acknowledge some limitations of this study. First, because of the correlational nature of this research, we cannot make causal inferences. Future research in this research field could adopt experimental designs to investigate whether different positive body image dimensions are causally linked with psychological wellbeing and self-acceptance. Moreover, we used a convenience sample, so our findings are not generalizable to the entire population. While utilizing social media for participant sampling offers both advantages and limitations, we endeavored to utilize various platforms to reach a diverse participant pool. However, it is important to recognize that individuals not active on social media may have been missed. Despite this limitation, our inclusive approach aimed to involve a broad range of participants through indirect means, like encouraging participants to disseminate the survey link within their social circles. Future research could benefit from employing a mixed-methods approach, combining social media sampling with traditional methods, and exploring alternative strategies like targeted advertising or partnerships with community organizations to enhance participant diversity and inclusivity. Moreover, previous research has also found that the relationships between variables outlined in sociocultural models of body image vary depending upon the sociocultural setting of the participants. For example, cultural specificities are important factors to consider when devising and analyzing sociocultural models of body image, with the focus on appearance potentially varying between countries [82,83]. Further attention is warranted to extrapolate how the effects of positive body image can vary depending on an individual’s sociocultural environment.

## 5. Conclusions

Our findings have relevant practical implications.

First, developing targeted interventions to enhance positive body image in its different facets might have a significant effect on the psychological wellbeing of both women and men psychological functioning by reducing the likelihood of maladaptive coping strategies. These interventions can be customized to address gender-specific needs and challenges, maximizing their effectiveness in promoting a healthy relationship with one’s body. Future research should focus on identifying the most effective components of interventions that enhance positive body image and exploring the mechanisms through which these interventions operate to optimize their impact across different populations. Longitudinal studies can provide insights into the sustained benefits of these interventions on individuals’ body image and psychological wellbeing, contributing to the development of comprehensive approaches to support greater wellbeing and resilience.

Furthermore, fostering men’s self-compassion towards their bodies can help them focus on their strengths and resources, leading to improved self-acceptance and realization of their true potential. Tailoring interventions to promote self-compassion among men, addressing gender-specific barriers and providing tools to cultivate self-compassion can support men in enhancing their overall wellbeing. Future research should investigate effective approaches to foster self-compassion in men and understand how these interventions impact men’s self-perception and potential. Longitudinal studies can examine the long-term effects of self-compassion interventions on men’s wellbeing and personal development, offering insights into maximizing the lasting benefits of self-compassion practices to empower men to lead fulfilling lives. Advancing knowledge in this area can lead to the development of interventions that help men embrace self-compassion and reach their full potential.

## Figures and Tables

**Figure 1 behavsci-14-00378-f001:**
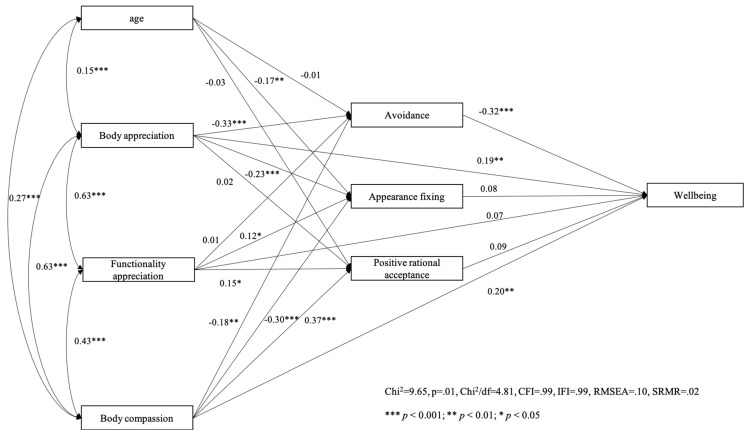
Women’s model with wellbeing as the outcome variable.

**Figure 2 behavsci-14-00378-f002:**
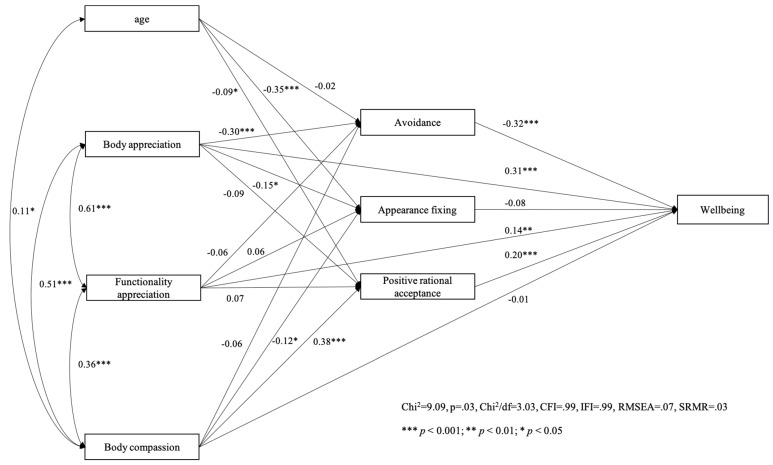
Men’s model with wellbeing as the outcome variable.

**Figure 3 behavsci-14-00378-f003:**
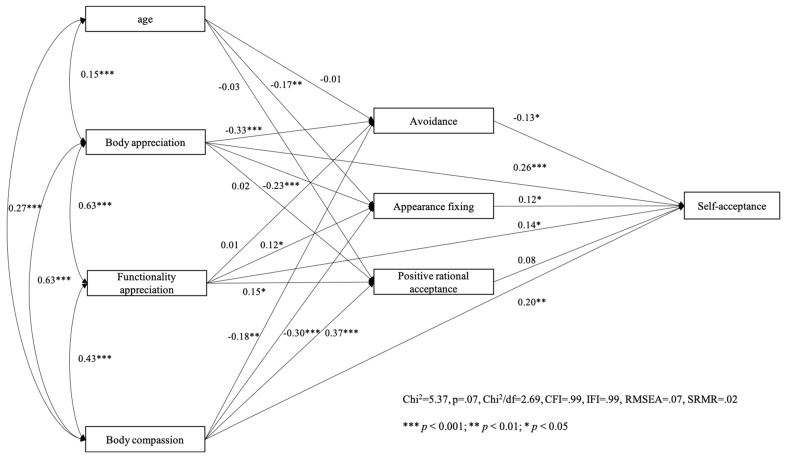
Women’s model with self-acceptance as the outcome variable.

**Figure 4 behavsci-14-00378-f004:**
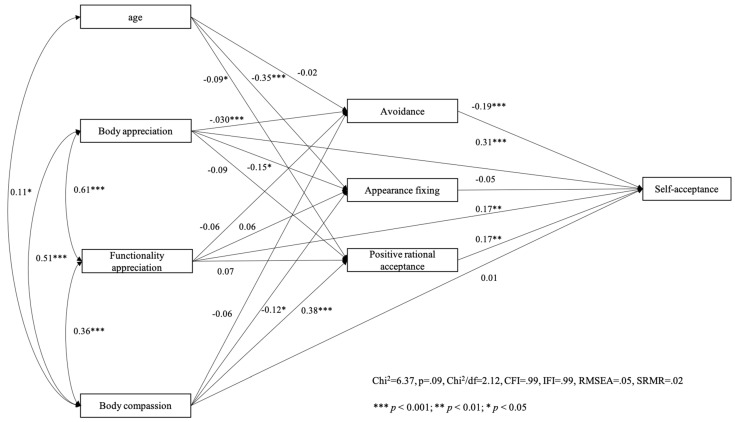
Men’s model with self-acceptance as the outcome variable.

**Table 1 behavsci-14-00378-t001:** Descriptive statistics.

	Men (*n* = 377)					Women (*n* = 372)				
	Min.	Max.	Mean (SD)	Asymmetry (SD)	Kurtosis (SD)	Min.	Max.	Mean (SD)	Asymmetry (SD)	Kurtosis (SD)
Body appreciation	1.10	5.00	3.74 (0.73)	−0.62 (0.13)	0.45 (0.25)	1.00	5.00	3.93 (0.71)	−1.07 (0.13)	2.20 (0.25)
Functionality appreciation	1.14	5.00	4.01 (0.68)	−1.28 (0.13)	2.68 (0.25)	1.10	5.00	3.41 (0.85)	−0.20 (0.13)	−0.38 (0.25)
Body Compassion	1.09	4.91	3.23 (0.59)	−0.22 (0.13)	1.09 (0.25)	1.00	4.96	3.08 (0.70)	−0.08 (0.13)	0.16 (0.25)
Appearance fixing	0.00	2.89	1.03 (0.69)	0.26 (0.13)	−0.42 (0.25)	0.00	3.00	1.40 (0.65)	−0.02 (0.13)	−0.36 (0.25)
Avoidance	0.00	2.57	0.67 (0.55)	0.82 (0.13)	0.19 (0.25)	0.00	3.00	0.91 (0.64)	0.57 (0.13)	−0.23 (0.25)
Positive rational acceptance	0.00	3.00	1.37 (0.62)	−0.07 (0.13)	−0.40 (0.25)	0.00	3.00	1.46 (0.58)	−0.09 (0.13)	0.03 (0.25)
Self-acceptance	0.33	6.00	3.70 (1.13)	−0.35 (0.13)	0.02 (0.25)	1.00	6.00	3.70 (1.11)	−0.15 (0.13)	−0.42 (0.25)
Overall Psychological Wellbeing	1.56	6.00	4.08 (0.78)	0.08 (0.13)	−0.31 (0.25)	2.06	5.89	4.01 (0.75)	0.07 (0.13)	−0.30 (0.25)

**Table 2 behavsci-14-00378-t002:** Intercorrelations between variables. Women’s results (*n* = 372) are reported below the diagonal. Men’s results (*n* = 377) are reported above the diagonal.

	1	2	3	4	5	6	7	8	9	10
1. BMI	−	0.39 ***	−0.06	−0.07	−0.02	−0.15 **	0.12 *	0	0.06	0.08
2. Age	0.19 ***	−	0.09	0	0.15 **	−0.39 ***	−0.06	−0.05	0.11 *	0.15 **
3. Body Appreciation	−0.18 ***	0.18 ***	−	0.61 ***	0.52 ***	−0.20 ***	−0.37 ***	0.15 **	0.52 ***	0.55 ***
4. Functionality appreciation	−0.20 ***	0.05	0.64 ***	−	0.36 ***	−0.07	−0.27 ***	0.16 **	0.44 ***	0.45 ***
5. Body Compassion	−0.06	0.28 ***	0.63 ***	0.44 ***	−	−0.23 ***	−0.24 ***	0.35 ***	0.34 ***	0.37 ***
6. Appearance fixing	0.03	−0.30 ***	−0.37 ***	−0.16 **	−0.44 ***	−	0.41 ***	0.34 ***	−0.15 **	−0.21 ***
7. Avoidance	0.26 ***	−0.12 *	−0.44 ***	−0.28 ***	−0.39 ***	0.48 ***	−	0.26 ***	−0.33 ***	−0.45 ***
8. Positive rational acceptance	0	0.08	0.34 ***	0.32 ***	0.43 ***	0.1	0.09	−	0.18 ***	0.16 **
9. Self-acceptance	0.02	0.18 ***	0.52 ***	0.43 ***	0.46 ***	−0.14 **	−0.30 ***	0.30 ***	−	0.83 ***
10. Overall psychological wellbeing	−0.02	0.23 ***	0.51 ***	0.39 ***	0.48 ***	−0.24 ***	−0.46 ***	0.25 ***	0.82 ***	−

Note. *** *p* < 0.001; ** *p* < 0.01; * *p* < 0.05.

**Table 3 behavsci-14-00378-t003:** Indirect effects (women).

Indirect Path	95% Confidence Interval for Self-Acceptance	95% Confidence Interval for Wellbeing
Body appreciation	−0.027; 0.065	0.043; 0.151
Body functionality	0.000; 0.060	−0.015; 0.061
Body compassion	−0.047; 0.084	−0.006; 0.139

Note. Bias-corrected bootstrap confidence intervals (CIs) derived from 5.000 bootstrap resamples were estimated to test for the significance of conditional indirect effects. The effects are considered significant if the CI values do not include zero.

**Table 4 behavsci-14-00378-t004:** Indirect effects (men).

Indirect Path	95% Confidence Interval for Self-Acceptance	95% Confidence Interval for Wellbeing
Body appreciation	0.014; 0.090	0.050; 0.138
Body functionality	−0.004; 0.052	−0.006; 0.070
Body compassion	0.027; 0.132	0.053; 0.162

Note. Bias-corrected bootstrap confidence intervals (CIs) derived from 5.000 bootstrap resamples were estimated to test for the significance of conditional indirect effects. The effects are considered significant if the CI values do not include zero.

## Data Availability

The data that support the findings of this study are available on request from the corresponding author. The data are not publicly available due to their containing information that could compromise the privacy of research participants. However, should anyone need to access a specific part of the datasets, we will do our best to comply with their request.

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
