# Peer review of "Positive Body Image and Psychological Wellbeing among Women and Men: The Mediating Role of Body Image Coping Strategies"

_behavsci, 2024, doi:10.3390/bs14050378_

Round 1
Reviewer 1 Report
Comments and Suggestions for Authors
In this manuscript, the authors investigated the role of body image coping strategies affecting the relationship between positive body image and personal wellbeing, also highlight the difference between genders.
The data are detailed presented, results and interpretation are well discussed.
Sincerely.
Reviewer 2 Report
Comments and Suggestions for Authors
This is a very clear and well developed article. The rationale for the study and gap in research is well made and there is significant exploration of literature to support study. Most literature is recent and relevant. Discussion and conclusion are where developed and data used to support key outcomes well.
Findings are analysed appropriately and are used effectively to respond to hypothesis.
Any drawbacks to using social media to sample participants? Might some 'types' be missed or was the range of social media sufficient to gain a range of participants?
Minor issues below
Line 81 and 84
Body image and coping strategies are key elements for one’s wellbeing. Indeed, search has shown that adopting certain coping strategies might be related to specific mental health outcomes [14, 15, 16, 17, 18]. This section could be elaborated on- what mental health outcomes- so it signposts to later sections of writing.
Line 84 body-image quality of life - not sure why body image quality of life mean? is a comma missing?
I enjoyed reading this work and felt the study was built on a very strong knowledge base and previous research. The genders being explored is useful and allows for interesting nuances to be explored.
Reviewer 3 Report
Comments and Suggestions for Authors
The article addresses the interesting issue of positive body image and its relationship to well-being and self-esteem with a focus on body image coping strategies. The presentation of research results is preceded by an analysis of theoretical constructs and a demonstration of the state of knowledge in the subject. The description of the methodology is correct. The reader is provided with the necessary knowledge about the subjects, the tests, the procedure of testing and analyzing the data. The results are presented coherently, consistently according to the adopted scheme. The discussion includes reference to the hypotheses and to information from the theoretical introduction. The authors are aware of the limitations to the results. The entire research project is coherent and consistently conducted.
Reviewer 4 Report
Comments and Suggestions for Authors
Dear Authors,
I think the article 'Positive body image and psychological well-being in women and men: the mediating role of body image coping strategies' is a valuable contribution to stress coping research.
Appearance in general and body image in particular have become very important constructs in today's society. In almost every aspect of our lives - on billboards, in shop windows, in every magazine, in ordinary conversations, in the amount of money, time and effort spent in the pursuit of 'beauty' - there is a strong emphasis on appearance. Because it is impossible to fully define what beauty actually is, beauty is put in inverted commas. I agree with you that the results of the research carried out fill a gap in this regard.
I think the article is worthy of publication in Behavioural Sciences, but it needs about a dozen corrections before publication, a list of which I include below.
Compared to the other sections, the introduction seems too long.
It is written in a very convoluted style which gives the reader the feeling of reading the same thing repeatedly. I think this important topic could be presented more clearly by cutting the introduction.
Specific comments on the text:
Line 32 - e.g. the abbreviation should be placed before the cited reference.
Lines 161-162, the numbering of the citations should be in a different order, e.g. [32, 40, 41].
Line 163 same remark
Materials and Methods
Line 247 - I have a dubst, how the percentage of married or cohabiting people was calculated, whether it was the total number of respondents or just those in a relationship, the author should explain this.
Throughout the text and tables, the way numerical values are expressed with decimals should be standardised. The authors are not consistent in this. Cronbach's coefficient values should also be expressed as present values with a 0 in front, as required by the journal.
Therefore,
Line 273 gives alpha values women: α=0.94, men: α=0.92
Line 280 - women: α=0.89, men: α=0.86
Lines 287-288 - women: α=0.87, men: α=0.88
Lines 305-307: (Women: Appearance fixation: α=0.85, avoidance: α=0.79; positive rational acceptance: α=0.78; men: Appearance fixation: = α = .85, avoidance: α = = .76, positive rational acceptance: α = .80). Should be: Women: Appearance fixing: α = 0.85, avoidance: α = 0.79; positive rational acceptance: α = 0.78; men: Appearance fixation: = α = 0.85, avoidance: α = = 0.76, positive rational acceptance: α = 0.80).
Lines 312-317 - is: n (women: α = .86; men: α = .86). High scores indicate high levels of wellbeing. As outlined in the hypotheses, in addition to the total score of the scale, we focused on the self-acceptance subscale, which consists of three items (e.g. 'When I look at the story of my life, I am satisfied with the way things have turned out'). The internal consistency for the self-acceptance subscale was good (women α = .80; men α = .82).
Line 335 - we interpret the Cronbach coefficient values differently, see DOI 10.1007/s11165-016-9602-2
Lines 444-446: phrase appears to be part of the template, should be removed.
Table 2 - What the asterisks mean? should be explained below the table.
Improvements are also needed in Tables 3 and 4.
I would suggest including a description below the tables; placing the values next to each other without naming them in the header confuses the reader, especially as there is no explanation in the text as to what the values refer to.
With regard to the publisher's requirements, it is worth formulating the main conclusions of the article because, as the authors themselves point out, the article has practical implications.
After the suggested changes have been made, the article "Positive body image and psychological wellbeing among women and men: the mediating role of body image coping strategies" should be published by Behavioural Sciences.
Yours sincerely
Reviewer
Round 2
Reviewer 4 Report
Comments and Suggestions for Authors
Dear Authors,
Thank you for considering my comments and adjusting the manuscript text accordingly. I think the revised manuscript meets all the journal's requirements and I'm happy to recommend it for publication in Behavioural Sciences.
Best wishes
Reviewer